# Intracardiac Echo Versus Fluoroscopic Guidance for Pulsed Field Ablation: Single-Center Real-Life Study

**DOI:** 10.3390/biomedicines13051186

**Published:** 2025-05-13

**Authors:** Vivek Joseph Varughese, James Pollock, Chandler Richardson, Dominic Vacca, Hata Mujadzic, Sultan Siddique

**Affiliations:** 1Internal Medicine, Prisma Health Richland, Columbia, SC 29203, USA; james.pollock@prismahealth.org (J.P.); chandler.richardson@prismahealth.org (C.R.); dominic.vacca@prismahealth.org (D.V.); 2Prisma Health Cardiology, Columbia, SC 29203, USA; hata.mujadzic@prismahealth.org (H.M.); sultan.siddique@prismahealth.org (S.S.)

**Keywords:** atrial fibrillation, catheter ablation, pulsed field ablation, intracardiac echocardiogram, fluoroless ablation

## Abstract

**Background:** Pulsed field ablation (PFA) is a novel non-thermal modality for catheter ablation (CA) in atrial fibrillation (AF) and has been replacing traditional thermal modalities. There have been studies in the past comparing fluoroscopic (FL) versus intracardiac echocardiogram (ICE) guidance for thermal ablation modalities. However, there have not been studies that compare outcomes for PFA performed under ICE versus FL guidance. **Methods:** This study was designed in a longitudinal cross-sectional format. A total of 196 patients who underwent PFA for AF at Prisma Health Richland were selected for the retrospective analysis. Patients were divided into two groups: those who underwent PFA under FL guidance (103 patients) versus ICE guidance (93 patients). The recurrence of atrial arrhythmias in the six-month follow-up period was studied. Multivariate regression analysis was performed to assess the difference in the association of either modality with recurrence of atrial arrhythmias. Bayesian non-inferiority models were used to analyze the non-inferiority between the modalities. **Results:** A total of 31 patients (30.1%) in the fluoro group had documented atrial arrhythmias in the six months following ablation. While 23 patients (24.7%) in the ICE group had documented atrial arrhythmias in the six-month follow-up period. The recurrence of AF was noted in 22.3% (22 patients) in the fluoro group and 14% (13 patients) in the ICE group. After running the multivariate regression analysis models, PFA under fluoroscopic guidance did not differ from ICE guidance, in terms of the recurrent atrial arrhythmias in the six-month follow-up (Adjusted Odds Ratio: 0.964; 95% CI: 0.336–2.772). The fluoro and ICE groups also did not differ in terms of six-month atrial fibrillation recurrence (Adjusted Odds Ratio: 2.43; 95% CI: 0.649–9.19). Non-inferiority analysis with Bayesian model was carried out, comparing the fluoro group and the ICE group in terms of freedom from arrhythmias in the six-month follow-up, and no inferiority was proved (95% confidence interval: −0.18–0.053), with a 61.03% chance of ICE-guided PFA being superior to fluoro guidance in terms of recurrence free interval, but statistical significance was not reached. **Conclusions:** Mean fluoroscopic time in the FL guidance group was 15.9 min, while no radiation exposure was documented in the ICE group. CA performed under FL versus ICE guidance did not differ statistically in terms of six-month recurrence of atrial arrhythmias in general and AF in particular.

## 1. Introduction

Atrial fibrillation (AF) is one of the most sustained arrhythmias, with a rising global prevalence. AF prevalence in the United States was estimated to be 5.2 million in 2010, with an anticipated rise to 12.1 million in 2030. The increasing trend in prevalence is attributed to the aging of the population, rising tide of obesity, increased survival in cardiac pathologies, and increased detection [1]. Even with the current guideline-directed management strategies involving rate control and anticoagulation, there is a 5% risk of cardiac death in the AF population. Moreover, 35% to 50% of patients with atrial fibrillation who receive adequate anticoagulation either receive inpatient therapy or die within 5 years. Trials like the EAST-AFNET4 [2] have proved that early rhythm control strategy decreased the risk of adverse cardiovascular outcomes in patients with early atrial fibrillation and concomitant cardiovascular pathologies. Electrical or pharmacological cardioversion can be an effective management strategy for rhythm control in newly onset AF; after considering the duration of symptoms, there is a need for preprocedural anticoagulation or cardiac imaging to rule out left atrial thrombus. Medications like amiodarone, sotalol, flecainide, etc., are utilized for the long-term rhythm management strategies in AF. As per the current AHA/ACC/HRS guidelines for the management of AF [1], catheter ablation (CA) holds a class I recommendation as a rhythm control strategy when antiarrhythmic medications have failed, are contraindicated or not desired, and present an ongoing need for rhythm control. There is also a class I recommendation for CA as a rhythm control strategy in paroxysmal AF in younger patients with fewer comorbidities, as well as in high AF symptom burden situations. And in patients undergoing CA, pulmonary vein isolation (PVI) is recommended as the primary lesion set unless a different specific trigger is identified.

Pulsed field ablation (PFA), as a novel energy source for cardiac ablation, has been shown to be tissue selective and is expected to decrease damage to non-cardiac tissue while providing high efficacy in pulmonary vein isolation [3]. PFA have been replacing the traditional thermal ablation strategies (radiofrequency, cryoballoon, etc.) due to its higher ease and durability at PVI and decreased risk of complications such as esophageal injuries, phrenic nerve paralysis, and pulmonary vein stenosis compared to traditional thermal strategies. Multiple trials like the PEFCAT [4], PEFCAT II, and IMPULSE have proved the efficacy of this ablation strategy. Catheter ablations have traditionally been performed under fluoroscopic guidance for catheter placement [5]. It is well established that radiation exposure increases the lifetime risk of malignancies, genetic defects, skin injuries, and cataracts [6]. Prolonged fluoroscopy during radiofrequency ablation may potentially cause a small increase in the lifetime risk of fatal malignancy, with lung malignancy being most likely. The estimated total lifetime excess risk of a fatal malignancy was 294 per million cases (0.03%) per 60 min of fluoroscopy associated with radiofrequency ablation. This risk will be higher if fluoroscopy units require higher output. The lungs are exposed to the maximum amount of radiation and are the most likely site of malignancy. The risk of radiation induced malignancy increases with an increase in the body mass index of patients [7]. The duration of ablation procedures is variable depending on several factors, but, on average, the effective radiation dose was approximately 15 mSv, or the equivalent of 150 chest radiographs, in one study [8]. Other studies have estimated the cumulative radiation burden from 53 to 60 min of fluoroscopy during ablation procedures results in 0.7 to 1.4 fatal malignancies per 1000 women and 1.0 to 2.6 per 1000 men [9]. Although the total fluoroscopy time is less accurate as a predictor of radiation-related malignancies when compared with the total radiation dose, strategies to decrease fluoroscopy time should, nonetheless, be pursued in any laboratory seeking optimal safety for its users, including patients, physicians, and laboratory staff. Fluoroless catheter ablation of all endocardial cardiac arrhythmias is feasible using current, and often standard, electrophysiology laboratory equipment. Nonfluoroscopic or “fluoroless” catheter ablation can be safely and effectively performed in adults with a variety of arrhythmias by only relying on intracardiac electrograms, electroanatomic mapping (EAM), and, in most instances, intracardiac echocardiography (ICE) for catheter guidance. ICE permits real-time visualization of catheters and cardiac structures, monitoring for procedural complication(s) (pericardial effusion), and direct visualization of structures that are not identifiable on fluoroscopy or the three-dimensional EAM (electroanatomic clinical map). ICE also allows operators to monitor ablation progression by edema formation and bubble formation as a predictor of steam pop and tissue–catheter contacts. There have been studies in the past comparing fluoroscopy versus ICE guidance in CA but mostly involving RFA.

ICE-guided PFA, by placing the ICE catheter in the left atrium to maintain direct visualization, has added advantages of ensuring direct contact with the tissue, maintaining the coaxiality of the PFA catheter with the veins, etc. Also based on ICE guidance, we can expand the basket size according to the size of the pulmonary vein anatomy (funnel shaped, bifurcation, anomalous vein, extra middle vein, etc.) While X-rays cannot give us PFA catheter coaxiality information or anomalous vein or extra vein location information, they can give us a method of indirect contact against pulmonary veins by looking at how the petals are moving, but it cannot give information if we are in contact with the non-pulmonary vein areas, e.g., posterior wall, roof, mitral isthmus, etc.

In our study, we aimed to compare the outcomes of PFA for AF carried out under fluoroscopy versus intracardiac ECHO (ICE) guidance in terms of efficacy, determined by a 6-month recurrence-free interval, and safety outcomes based on procedure-related complications.

## 2. Materials and Methods

### 2.1. Study Design

A total of 196 patients undergoing PFA for AF at Prisma Health Richland were selected for this study. Inclusion criteria were adults over the age of 18 who underwent successful PVI using PFA. Patients were subdivided into the fluoroscopic group (103 patients) and intracardiac ECHO (ICE) group (93 patients). The primary efficacy endpoint was determined by a six-month symptom-free interval after ablation. Criteria for determining recurrence were defined as objective atrial arrhythmias (atrial fibrillation, atrial flutter, atrial tachycardia, and bradycardia) detected using a loop recorder, cardiac event monitor, pacemaker, or the 12 lead ECG obtained during office or emergency room visits. AF detected with duration >30 s was the additional criteria used to determine symptom recurrence. The requirement of repeat cardioversion or additional ablation in the six-month follow-up was also used as a criterion for symptomatic recurrence. Secondary safety endpoints were determined by the occurrence of procedural complications as well emergency room visits in the 30 days following CA for complications related to the procedure (excluding recurrence of arrhythmias).

Three independent operators carried out the ablations. Of the 103 patients who underwent PFA under fluoroscopic guidance, electroanatomic mapping was conducted using Bio sense the Webster mapping catheter in 76 patients and the OCTARAY mapping system in 26 patients. Electroanatomic mapping was performed using the ESI mapping system in the 93 patients who underwent PFA under ICE guidance.

### 2.2. Methods

The normality of each variable was assessed using the Kolmogorov–Smirnov test. Quantitative data were expressed by mean standard deviation, and the differences between the means of groups were tested with the unpaired *t*-test. Quantitative data were expressed in percentages, and the differences between the two groups were tested using the chi-square test or Fisher exact test. Multivariate logistic regression analysis was performed to analyze the factors associated with recurrent atrial arrhythmias and recurrent atrial fibrillation specifically. Age, gender, left atrial dimension (as a surrogate marker for duration of atrial fibrillation), OSA (with and without CPAP use), type of atrial fibrillation (paroxysmal versus persistent/chronic), heart failure, use of anti-arrhythmic medications post ablation, alcohol abuse, and pulmonary hypertension were used in the regression analysis as these are factors with proven association with severity in atrial fibrillation. In addition, smoking, presence of mitral regurgitation, and presence of aortic regurgitation were included in the regression analysis as they showed significant variance in the one-way ANOVA. The Adjusted Odds Ratio for association was calculated with a 95% confidence interval. A ‘*p*’ value less than 0.05 was used to determine the statistical significance. The Bayesian non-inferiority model with a 95% confidence interval was used to determine the non-inferiority between the two groups. Initial data entry was performed in the Redcap system, and further statistical analysis was performed using the SPSS-25 version.

## 3. Results

The baseline patient characteristics and outcomes are summarized in Table 1 and Figure 1, respectively. The mean age and the mean left atrial (LA) dimension did not differ between the fluoro and ICE group. The mean age of patients included in the fluoro group was 68.65, while that in the ICE group was 67.57. The mean LA dimension was 4.03 ± 0.66 cm in the fluoro group and 4.09 ± 0.64 cm in the ICE group. Gender stratification also proved no statistically significant difference between both the groups. The patient groups did not differ in the type of AF, i.e., paroxysmal, persistent, or chronic. The mean CHADSVASC score in the fluoro group was 3.35 ± 1.48, while, in the ICE group, it was 3.05 ± 1. Antiarrhythmic medications after ablation were required in 74 patients. This amounted to 47.3% (44 patients) in the ICE group, while it was noted to be 35% (36 patients) in the fluoro group. A total of 13 patients (12.6%) in the fluoro group underwent watchman placements in the six-month follow-up period, while 5 patients (5.4%) in the ICE group underwent watchman placement in the follow-up period.

Comparing the prevalence of comorbid conditions in the groups, 19 patients (18.4%) in the fluoro group had documented diagnosis of heart failure with reduced ejection fraction (HFrEF), and 15 patients (16.1%) in the ICE group had documented HFrEF. The prevalence of obesity was noted to be at 49.5% (51 patients) in the fluoro group and 61.3% (57 patients) in the ICE group. A total of 34 patients (33%) in the fluoro group used CPAP for their obstructive sleep apnea (OSA), while 29 patients (31.2%) in the ICE group used CPAP for their OSA. Additionally, 21 patients (20.4%) in the fluoro group did not use CPAP for their documented OSA, and the number was 17 patients (18.3%) in the ICE group. Mitral stenosis (MS) was documented in two patients (2.2%) in the ICE group, while no patients in the fluoro group had documented MS. The prevalence of alcohol abuse was 20.4% (21 patients) in the fluoro group and 7.5% (7 patients) in the ICE group. The prevalence of smoking was 10.7% (11 patients) in the fluoro group and 6.5% (6 patients) in the ICE group. Patients in the fluoro group had a statistically significant higher prevalence of alcohol abuse compared to the ICE group. The prevalence of mitral as well as aortic regurgitation was found to be significantly higher in the ICE group.

Considering the technical factors associated with the ablation procedures in the groups, 52.42% (54 patients) in the fluoro group had additional ablations (atrial flutter, SVT, CTI, etc.) along with the PFA of PV, while 67.77% (63 patients) in the ICE group had additional ablations along with PVI. The mean fluoro time was 15.90 (±5.17) minutes in the fluoro group. Total procedure time was not statistically different between the groups. The mean of the total procedure time was noted to be 98.75 ± 34.16 min in the fluoro group, while it was 103.75 ± 45.75 min in the ICE group. ESI mapping system was used for all patients in the ICE group, while Biosense Webster CARTO 3D was used in 76 patients (73.7%), and OCTARAY system was used in 26 patients (25.24%) in the fluoro group.

### 3.1. Primary Efficacy Endpoints

Primary efficacy endpoint was defined by a six-month symptom-free interval post ablation. The recurrence of atrial arrhythmias was to be detected objectively either from a loop recorder, event monitor, pacemaker, or 12 lead EKG. Atrial fibrillation, atrial flutter, atrial tachycardia, and bradyarrhythmia were considered recurrent atrial arrhythmias. A total of 31 patients (30.1%) in the fluoro group had documented atrial arrhythmias in the six months following ablation. On the other hand, 23 patients (24.7%) in the ICE group had documented atrial arrhythmias in the six-month follow-up period. The primary safety endpoint was reached in 69.9% (72 patients) in the fluoro group and 75.3% (70 patients) in the ICE group. The recurrence of AF was noted in 22.3% (22 patients) in the fluoro group, and 14% (13 patients) in the ICE group. The recurrence of other atrial arrhythmias is summarized in Table 1. Cardioversion for recurrent atrial fibrillation was required in 8.7% (nine patients) in the fluoro group and 8.6% (eight patients) in the ICE group. Repeat ablation and PVI was required in 2.9% (three patients) in the fluoro group, and only one patient (1.1%) in the ICE group required repeat ablation and PVI in the six-month follow-up period. The results are summarized in Figure 2.

After running the multivariate regression analysis models (Table 2), PFA under fluoroscopic guidance did not differ from PFA under ICE guidance, in terms of the primary efficacy endpoint in the absence of recurrent atrial arrhythmias in the six-month follow-up (Adjusted Odds Ratio: 0.964). The fluoro and ICE groups also did not differ in terms of six-month atrial fibrillation recurrence (Table 3), after multivariate regression analysis (Adjusted Odds Ratio: 2.43; 95% CI: 0.649–9.19). After running regression analysis models, smoking was the only associated comorbidity that had significant association with recurrent atrial arrhythmias in the six months following ablation (Adjusted Odds Ratio: 11.54; 95% CI: 1.104–120.72; *p* value: 0.041). Non-inferiority analysis with the Bayesian model was performed, comparing the fluoro group and ICE group in terms of freedom from arrhythmias in the six-month follow-up, and no inferiority was proved (95% confidence interval: −0.18–0.053), with a 61.03% chance of ICE-guided PFA being superior to fluoro guidance in terms of the recurrence-free interval, but statistical significance was not reached.

### 3.2. Secondary Safety Endpoints

Pericardial effusion and tamponade occurred in one patient (1.1%) in the ICE group. One patient in the fluoro group was admitted within the 30 days following ablation for dysphagia. The patient underwent symptomatic management, and no further diagnostic workup was documented. Two patients in the fluoro group and one patient in the ICE group had admissions for gastrointestinal bleeding (GIB) in the 30 days following ablation. No periprocedural mortality was noted in either of the groups.

## 4. Discussion

Three-dimensional echocardiography allows imaging and analysis of cardiovascular structures as they move in time and space, thus creating the possibility for the creation of 4D datasets (3D + time). Intracardiac echocardiography (ICE) further broadens the spectrum of echocardiographic techniques by allowing detailed imaging of intracardiac anatomy with 3D reconstructions. ICE can also potentially eliminate the need for preprocedural TEE for the exclusion of LAA thrombus and CT/MRI required to detect potential anatomic variations. In the retrospective analysis performed by Sriram et al. [10], 122 patients who had a negative left atrial thrombus, as per transesophageal echocardiogram, had a re-evaluation of the left atrium for thrombus using ICE. ICE was noted to have a complimentary value in re-screening the LA/LAA for thrombus after a recent negative or equivocal TEE. The presence of SEC during TEE increases the probability of finding a thrombus with ICE, which could potentially be dislodged during catheter manipulation. The right ventricular inflow tract (RVIT) was the best ICE view for detecting left atrial thrombus based on the results of this retrospective analysis. Despite the ubiquity of these tools, most operators continue to use fluoroscopy during AF ablation, perhaps due to habit, as most electrophysiologists have been trained to perform the procedure in this fashion. More importantly, most interventional electrophysiologists are not trained, or they feel uncomfortable manipulating the ICE catheter in the left atrium. However, operators may also be reluctant to make the transition to fluoroless AF ablation as there has been a general paucity of data regarding its safety and long-term effectiveness [11].

There have been several studies in the past comparing the efficacy and safety between fluoroscopy versus ICE guidance for CA in AF, but the catheter ablation strategies mainly involved thermal modalities like RFA. Lyan et al. [12] conducted a retrospective analysis on 481 patients undergoing RFA for PVI, where 245 patients underwent RFA using ICE guidance and 236 patients underwent conventional fluoroscopic guidance. Both groups were found to not differ in the primary efficacy endpoint of the 12-month symptom-free interval. Total procedural time was also not found to be different in the two groups. In another retrospective analysis comparing the two guidance modalities by Lurie et al. [11], ICE modalities were found to have similar safety and efficacy endpoints as the traditional fluoroscopic approach. In the ADVENT trial [13], which was a head-to-head comparison between RFA and PFA in drug refractory AF, PFA was proved to be superior to RFA in terms of the primary efficacy endpoint of symptom-free interval, but one specific point observed in the PFA group was the higher fluoroscopic time and radiation exposure involved. In our analysis, the mean total procedure time was noted to be 98.75 min in the fluoro group and 103.75 min in the ICE group. The total procedure time did not differ between the PFA performed with either of the two guidance modalities (*p* value: 0.38). The mean fluoro time exposure was 15.90 ± 5.17 in the fluoro group. The radiation exposure and fluoro time was zero in the ICE group as no fluoroscopic guidance was utilized in either the trans-septal puncture or catheter placement in this group. However, investigators of the ADVENT trial had not specified the guidance modality employed for patients in the PFA and thermal ablation group.

Regarding the primary efficacy endpoint of the six-month symptom-free interval, both groups did not differ, be it in terms of recurrent atrial arrhythmias in the six-month follow-up or AF recurrence in particular. We compared the results with the findings in the AdmIRE Pivotal trial [14], which was a large clinical trial analyzing outcomes in 277 patients that underwent PFA. Per the investigators’ recommendations, 66 patients in the study group underwent PFA under ICE guidance. The total procedure time was noted to be 90 min (65–119), and the mean fluoroscopic time was 7.1 min. Three patients had pericardial tamponade related to the procedure, but the investigators had not specified the guidance modality employed in these patients. Primary effectiveness outcomes (defined by freedom from recurrence in the 12-month follow-up after a 3-month blanking period) were similar between patients treated without fluoroscopy (48 of 66; 72.7% [95% CI, 60.4–83.0%]) and >0 min fluoroscopy (135 of 180; 75.0% [95% CI, 68.0–81.1%]; *p* = 0.74). In our study, after running regression analysis models on factors associated with the recurrence of atrial arrhythmias in the six-month follow-up period, smoking was the only factor found to have a significant association with the recurrence of atrial arrhythmias (Adjusted Odds Ratio: 11.53; 95% CI: 1.10–121.71; *p* value < 0.00). Both the fluoro and ICE groups did not differ in terms of association with the recurrence of atrial arrhythmias in general or atrial fibrillation. When running the non-inferiority models, PFA performed under ICE guidance was noted to have a 61.03% chance of being superior to fluoro-guided ablation in terms of recurrence-free interval, but it did not reach statistical significance (95% CI: −0.18–0.053). In another multicenter study analyzing 667 patients that underwent PFA under ICE guidance for pulmonary vein isolation, 100% pulmonary vein isolation was achieved. Only one patient had permanent phrenic nerve damage documented [15].

Considering the safety endpoints, one case of procedure-related pericardial tamponade was observed in the ICE group. Although esophageal injury is a known complication following thermal modalities of ablation [16], the risk is lower in PFA. Only one patient who underwent PFA under fluoro guidance was admitted in the 30 days following the procedure for dysphagia, but further endoscopic workups were not performed, and the patient was managed symptomatically.

## 5. Conclusions

Pulsed field ablation (PFA) did not differ in terms of the six-month recurrence of atrial arrhythmias based on the guidance chosen, i.e., fluoroscopic versus intracardiac ECHO (ICE). The total procedure times did not differ statistically in both groups. The mean fluoro time was 15.90 ± 5.17 in the fluoro group, while no fluoroscopy was utilized for PFA using ICE guidance.

## Figures and Tables

**Figure 1 biomedicines-13-01186-f001:**
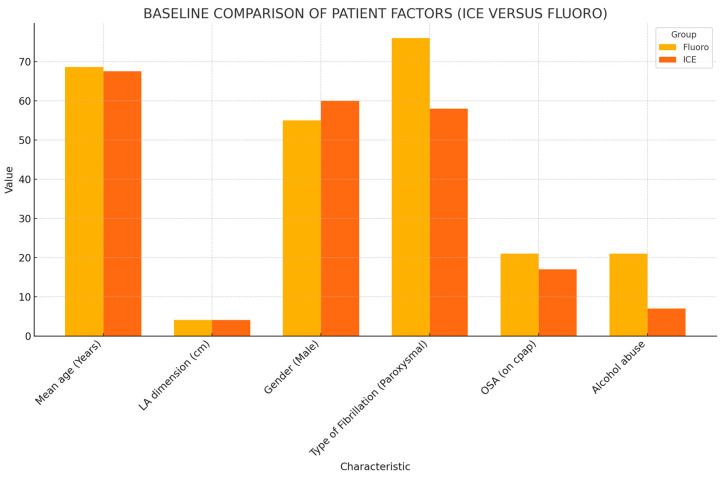
Baseline comparison of patient factors.

**Figure 2 biomedicines-13-01186-f002:**
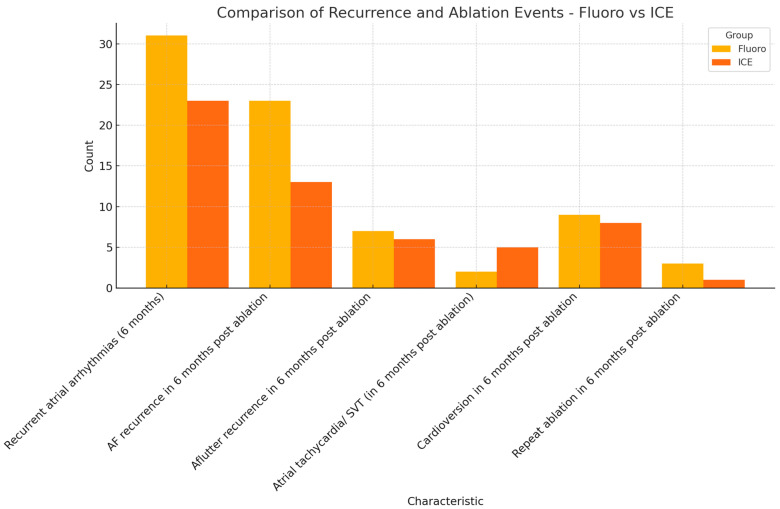
Comparison of primary efficacy endpoints (fluoro versus ICE).

**Table 1 biomedicines-13-01186-t001:** Comparison of baseline characteristics between fluoro and ICE groups.

		Fluoro (n = 103)	Intracardiac ECHO Group (n = 93)	*p* Value
Mean age (Years)		68.65 ± 10.49	67.57 ± 10.43	0.47
LA dimension (cm)		4.03 ± 0.66	4.09 ± 0.64	0.54
Fluoro time (mins)		15.90 ± 5.17	0	<0.01
Total procedure time (min)		98.75 ± 34.16	103.75 ± 45.75	0.38
Additional ablation (accessory tracts)		54 (52.42%)	63 (67.77%)	0.12
Gender	Male	55 (53.4%)	60 (64.5%)	0.11
	Female	48 (46.6%)	33 (35.5%)	
Type of fibrillation	Paroxysmal	76 (73.8%)	58 (62.4%)	0.10
	Persistent	27 (26.2%)	33 (35.5%)	
	Chronic	0 (0.0%)	2 (2.2%)	
Antiarrhythmic med use (post ablation)	No	67 (65.0%)	49 (52.7%)	0.07
	Yes	36 (35.0%)	44 (47.3%)	
Recurrent atrial arrhythmias (6 months)	No	72 (69.9%)	70 (75.3%)	0.40
	Yes	31 (30.1%)	23 (24.7%)	
AF recurrence in 6 months post ablation	No	80 (77.7%)	80 (86.0%)	0.13
	Yes	23 (22.3%)	13 (14.0%)	
Aflutter recurrence in 6 months post ablation	No	96 (93.2%)	87 (93.5%)	1.0
	Yes	7 (6.8%)	6 (6.5%)	
Atrial tachycardia/SVT ( in 6 months post ablation)	No	100 (97.1%)	88 (94.6%)	0.20
	Yes	2 (1.9%)	5 (5.4%)	
Cardioversion in 6 months post ablation	No	94 (91.3%)	85 (91.4%)	0.97
	Yes	9 (8.7%)	8 (8.6%)	
Repeat ablation in 6 months post ablation	No	100 (97.1%)	92 (98.9%)	0.62
	Yes	3 (2.9%)	1 (1.1%)	
Watchman placement	No	89 (86.4%)	88 (94.6%)	0.07
	Yes	13 (12.6%)	5 (5.4%)	
HFrEF (before ablation)	No	82 (79.6%)	71 (76.3%)	0.80
	Yes	19 (18.4%)	15 (16.1%)	
Obesity		51 (49.5%)	57 (61.3%)	0.08
OSA (on cpap)		34 (33.0%)	29 (31.2%)	0.76
OSA (not on cpap)		21 (20.4%)	17 (18.3%)	0.68
Pulmonary hypertension severity	Mild	27 (26.2%)	12 (12.9%)	0.15
	Mild to moderate	12 (11.7%)	0 (0.0%)	
	Moderate	17 (16.5%)	10 (10.8%)	
	Moderate to severe	3 (2.9%)	1 (1.1%)	
	Severe	2 (1.9%)	2 (2.2%)	
Alcohol abuse		21 (20.4%)	7 (7.5%)	0.01
Smoking		11 (10.7%)	6 (6.5%)	0.32
Mitral regurgitation		19 (18.4%)	51 (54.8%)	<0.001
Mitral stenosis		0 (0.0%)	2 (2.2%)	0.21
Aortic regurgitation		3 (2.9%)	23 (24.7%)	<0.001
Valve replacement mechanical/bioprosthetic		3 (2.9%)	3 (3.2%)	1.0
Mean CHADSVASC		3.35 ± 1.48	3.05 ± 1.74	0.20
Mapping catheter used		CARTO 3D: 76	ESI Mapping System: 93	
		OCTARAY:26		

**Table 2 biomedicines-13-01186-t002:** Multivariate logistic regression analysis for factors associated with recurrent atrial arrhythmias (6 months).

	*p* Value	Adjusted Odds Ratio	95% CI for Odds Ratio	
			Lower	Upper
Persistent fibrillation	0.555	0.728	0.253	2.091
Fluoro vs. intracardiac ECHO group	0.946	0.964	0.336	2.764
Obesity	0.643	1.273	0.459	3.536
OSA not on CPAP	0.481	0.654	0.200	2.133
Alcohol abuse	0.244	0.398	0.085	1.872
Smoking	0.041	11.543	1.104	120.727
Pulmonary hypertension severe	0.347	3.338	0.270	41.200

**Table 3 biomedicines-13-01186-t003:** Multivariate logistic regression analysis for factors associated with recurrent atrial fibrillation (6 months).

	*p* Value	Adjusted Odds Ratio	95% CI for Odds Ratio	
			Lower	Upper
Persistent fibrillation	0.977	1.018	0.315	3.291
Fluoro vs. intracardiac ECHO group	0.187	2.443	0.649	9.198
Obesity	0.916	1.065	0.332	3.420
OSA not on CPAP	0.479	1.560	0.455	5.340
Alcohol abuse	0.320	.415	0.073	2.354
Smoking	0.095	5.461	0.744	40.062
Pulmonary hypertension severe	0.139	6.873	0.534	88.414

## Data Availability

The original contributions presented in this study are included in the article. Further inquiries can be directed to the corresponding author.

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
