# Peer review of "Intracardiac Echo Versus Fluoroscopic Guidance for Pulsed Field Ablation: Single-Center Real-Life Study"

_biomedicines, 2025, doi:10.3390/biomedicines13051186_

Round 1
Reviewer 1 Report
Comments and Suggestions for Authors
I have no substantive comments on the manuscript.
A few points, though, I would like to clarify with the authors:
Since this study was not randomised and the groups had some differences initially, did the authors think to perform a correction for these differences (PSM?)? For example, Table 1 shows that the ECHO group had more frequent mitral and aortic regurgitation. Could this have affected the results? If these patients were excluded from the analysis, perhaps the rate of recurrent AF was significant?
If the regurgitation was not significant, then the atvors need to define what they thought regurgitation was in this context. Or specify the degrees.
Author Response
Comment: Since this study was not randomised and the groups had some differences initially, did the authors think to perform a correction for these differences (PSM?)? For example, Table 1 shows that the ECHO group had more frequent mitral and aortic regurgitation. Could this have affected the results? If these patients were excluded from the analysis, perhaps the rate of recurrent AF was significant?
If the regurgitation was not significant, then the authors need to define what they thought regurgitation was in this context. Or specify the degrees.
Response: Hi,
Thanks for reviewing our article. The methods section was described more in detail in the supplementary material sections, but we have included this in the discussion section for better clarity.
Factors included in the regression analysis:
-Age, Gender, Nature of afib ( paroxysmal versus persistent), anti arryhtmic med use, OSA, alcohol abuse, HFrEF, pulmonary hypertension: as these factors have proven association with adverse outcomes in afib
-variables with significant variance in one way ANOVA: MR, AR and smoking
MR and AR were not excluded, and were included in the regression analysis. We are continuing to collect patient data for the next 6 months to see if one year follow up would give a statistically significant result.
Only grade 3+/4+ MR were included ( mild -moderate MR was excluded). In case of AR, severe AR with LVESV < 5 cm or ERO > 0.3 cm2 were included
Reviewer 2 Report
Comments and Suggestions for Authors
Dear Sirs,
I would like to thank you very much for sending the review entitled: „Intracardiac Echo Versus Fluoroscopic Guidance for Pulsed Field Ablation: Single Center Real Life Study.”
Study analyzed retrospective 196 patients who underwent PFA for AF. Patients were divided into two groups: those who underwent PFA under FL guidance (103 patients) versus ICE guidance (93 patients). Recurrence of atrial arrhythmias in the six month follow up period was studied. Multivariate regression analysis was done to assess a difference in association of either modality with recurrence of atrial arrhythmias. Mean fluoroscopic time in the FL guidance group was 15.9 minutes, while no radiation exposure was documented in the ICE group. CA done under FL versus ICE guidance did not differ statistically in terms of six-month recurrence of atrial arrhythmias in general, and AF in particular.
This is a very interesting and necessary analysis. In particular, it highlights the need to optimise tissue contact in order to improve treatment efficacy. What is also important criteria for determining recurrence was defined as objective atrial arrhythmias (Atrial Fibrillation, Atrial Flutter, Atrial Tachycardia, Bradycardia) detected using loop recorder, cardiac event monitor, pacemaker, or by 12 lead ECG obtained during office or emergency room visits.
I believe the article will be of great interest to readers.
Yours sincerely,
Reviewer
Reviewer 3 Report
Comments and Suggestions for Authors
Congratulations to the authors for having provided new insight about such a relevant topic as the use of ICE in PFA; I just have some comments about the manuscript
Multivariate logistic regression was performed; however, the manuscript does not clearly explain the criteria used for variable selection.
Although key adverse events such as pericardial tamponade, gastrointestinal bleeding, and dysphagia are mentioned, no formal statistical comparison of safety outcomes was conducted; the results are presented descriptively only.
Was successful pulmonary vein isolation documented for every case immediately post-procedure?
Authors should include in order to empower their discussion the latest evidences about the fundamental role of the ICE in PFA Ablation (doi: 10.1016/j.jacep.2024.11.009)
The manuscript currently lacks any figures. Including a Kaplan-Meier survival curve illustrating arrhythmia-free survival over the six-month follow-up period would significantly enhance the manuscript's visual appeal and data presentation.
Author Response
Comment: Multivariate logistic regression was performed; however, the manuscript does not clearly explain the criteria used for variable selection.
Response: We have expanded the methods section explaining the variables included in the multivariate regression analysis: this was there in the supplementary index, hence was not included in the previous version.
Comment: Although key adverse events such as pericardial tamponade, gastrointestinal bleeding, and dysphagia are mentioned, no formal statistical comparison of safety outcomes was conducted; the results are presented descriptively only.
Was successful pulmonary vein isolation documented for every case immediately post-procedure?
Response: Yes, only procedures with successful PV isolation were included in the study. This is mentioned in the study design
Since the number of adverse events were minimal, we did not attempt statistical comparison
Comment: The manuscript currently lacks any figures. Including a Kaplan-Meier survival curve illustrating arrhythmia-free survival over the six-month follow-up period would significantly enhance the manuscript's visual appeal and data presentation.
Response: We did not have the time to event data, hence could not do the KM curve. However, we have added comparison bar charts to make the data more visually appealing
Comment: Authors should include in order to empower their discussion the latest evidences about the fundamental role of the ICE in PFA Ablation (doi: 10.1016/j.jacep.2024.11.009)
Response: doi: 10.1016/j.jacep.2024.11.009: this study has been added (reference 15)
Round 2
Reviewer 3 Report
Comments and Suggestions for Authors
My compliments to the authors for ther efforts in improving their paper.